# A Metabolomic Profile Predictive of New Osteoporosis or Sarcopenia Development

**DOI:** 10.3390/metabo11050278

**Published:** 2021-04-28

**Authors:** Kana Miyamoto, Akiyoshi Hirayama, Yuiko Sato, Satsuki Ikeda, Midori Maruyama, Tomoyoshi Soga, Masaru Tomita, Masaya Nakamura, Morio Matsumoto, Noriko Yoshimura, Takeshi Miyamoto

**Affiliations:** 1Department of Orthopedic Surgery, Kumamoto University, 1-1-1 Honjo, Chuo-ku, Kumamoto 860-8556, Japan; kana2001@galaxy.ocn.ne.jp; 2Institute for Advanced Biosciences, Keio University, 246-2 Mizukami, Kakuganji, Tsuruoka, Yamagata 997-0052, Japan; hirayama@ttck.keio.ac.jp (A.H.); satsuki@ttck.keio.ac.jp (S.I.); la52la.green@gmail.com (M.M.); soga@sfc.keio.ac.jp (T.S.); mt@sfc.keio.ac.jp (M.T.); 3Department of Orthopedic Surgery, School of Medicine, Keio University, 35 Shinano-machi, Shinjuku-ku, Tokyo 160-8582, Japan; bapybapy@hotmail.com (Y.S.); masa@keio.jp (M.N.); morio@a5.keio.jp (M.M.); 4Department of Advanced Therapy for Musculoskeletal Disorders II, Keio University School of Medicine, 35 Shinano-machi, Shinjuku-ku, Tokyo 160-8582, Japan; 5Department of Musculoskeletal Reconstruction and Regeneration Surgery, Keio University School of Medicine, 35 Shinano-machi, Shinjuku-ku, Tokyo 160-8582, Japan; 6Department of Preventive Medicine for Locomotive Organ Disorders, 22nd Century Medical and Research Center, The University of Tokyo, Hongo 7-3-1, Bunkyo-ku, Tokyo 113-8655, Japan; noripu@rc4.so-net.ne.jp

**Keywords:** osteoporosis, sarcopenia, metabolome, cohort study, the Research on Osteoarthritis/Osteoporosis Against Disability (ROAD)

## Abstract

The increasing number of patients with osteoporosis and sarcopenia is a global concern among countries with progressively aging societies. The high medical costs of treating those patients suggest that prevention rather than treatment is preferable. We enrolled 729 subjects who attended both the second and third surveys of the Research on Osteoarthritis/Osteoporosis Against Disability (ROAD) study. Blood samples were collected from subjects at the second survey, and then a comprehensive metabolomic analysis was performed. It was found that 35 had newly developed osteoporosis at the third survey performed four years later, and 39 were newly diagnosed with sarcopenia at the third survey. In the second survey, we found that serum Gly levels were significantly higher even after adjustment for age, sex, and BMI in subjects with newly developed osteoporosis relative to those who remained osteoporosis-negative during the four-year follow-up. We also show that serum taurine levels were significantly lower at the second survey, even after adjustment for age, sex, and BMI in subjects with newly developed sarcopenia during the four-year follow-up compared with those not diagnosed with sarcopenia at the second or third surveys. Though our sample size and odds ratios were small, increased Gly and decreased taurine levels were found to be predictive of new development of osteoporosis and sarcopenia, respectively, within four years.

## 1. Introduction

Osteoporosis and sarcopenia are the most important musculoskeletal diseases of the elderly [1]. Both develop with aging and are risks for loss of activity of daily living (ADL) and quality of life (QOL), as well as mortality, in the elderly [2,3]. As the both the number and rate in the elderly increase in various populations, preventing these diseases before they develop is mandatory. To do this, factors predictive of future disease are needed. However, both osteoporosis and sarcopenia are multifactorial and not attributable to a single factor, making prediction difficult.

Osteoporosis is characterized by reduced bone strength and increased risk for fragility fractures, and morbidity rates reportedly increase with age [4]. Known risks for osteoporosis include aging, menopause, smoking, excessive alcohol consumption, steroid use, genetic factors, and inflammatory or metabolic disease. Metabolites associated with low bone mass have been identified [5,6], but no longitudinal predictors have been reported. The fracture risks assessment tool (FRAX^®^) can predict fragility fractures based on responses to questionnaires relevant to age, sex, weight, height, previous fracture, parent fractured hip, current smoking and alcohol habits, glucocorticoid use, rheumatoid arthritis, and secondary osteoporosis; however, no biological markers are used in this assessment [7].

Sarcopenia is defined as a progressive loss of muscle mass with reduced muscle strength and/or function with age [8,9]. Known risks for sarcopenia include aging, immobilization, and cachexia [10]. The European Working Group on Sarcopenia in Older People (EWGSOP) developed a practical clinical definition and diagnostic criteria for sarcopenia in 2010 [8]. After that, the Asian Working Group for Sarcopenia (AWGS) provided appropriate diagnostic cutoff values for Asian populations [9]. Osteoporosis and sarcopenia frequently simultaneously develop in the elderly [11], suggesting that common predictors in addition to aging may exist for these diseases.

The present study was performed using Osteoarthritis/Osteoporosis Against Disability (ROAD) cohorts established in 2005. The ROAD study is a national, prospective study of musculoskeletal diseases that consists of population-based cohorts from several communities in Japan. Details of the cohort profiles have been reported elsewhere [4,12]. Briefly, between 2005 and 2007, a baseline database including the clinical and genetic information of 3040 residents (1061 men and 1979 women with a mean age of 70.3 years (standard deviation (SD): 11.0); 71.0 (10.7) years for men and 69.9 (11.2) years for women) was created. The study revealed that sarcopenia prevalence increases by 60 years of age and that four years after sarcopenia development, subjects become at risk for osteoporosis [11].

Metabolites are the end products of various metabolic pathways, and their levels overall reflect individuals’ metabolomic status [13,14]. Metabolomic analysis can detect changes or differences in metabolite levels and is thus useful to diagnose diseases such as cancer and to detect low bone mass or bone status in human and animals [15,16,17,18]. Since osteoporosis and sarcopenia are metabolic diseases that disrupt bone and muscle homeostasis, metabolomic analyses could predict diseases before development.

Here, we undertook the metabolomic analysis of serum samples collected at the second follow-up of ROAD and found that levels of the amino acid glycine (Gly) were significantly higher in subjects not identified as having osteoporosis at that second survey but who had newly developed osteoporosis during the four-year follow-up, compared with osteoporosis-negative subjects at the second and third surveys, after adjustment for age, sex, and BMI. Similarly, levels of taurine were significantly lower in subjects newly defined as exhibiting sarcopenia at the third survey after adjustment for age, sex, and BMI. Thus, Gly and taurine could serve as predictors of the future development of osteoporosis and sarcopenia, respectively.

## 2. Results

### 2.1. High Serum Gly Levels Are Significantly Associated with New Osteoporosis Development

Among study participants, 460 did not meet the osteoporosis criteria at the second survey and were thus judged osteoporosis-negative, but 35 (7 males and 28 females) of those 460 were newly defined as positive for osteoporosis at the third survey. We compared patient characteristics at the second survey between subjects evaluated at the third survey as new osteoporosis (new OP) or non-osteoporosis (non-OP) (Table 1). That analysis further indicated that the proportion of subjects either of relatively advanced age or female gender was significantly higher in new- versus non-OP groups (Table 1). In contrast, values indicative of BMD, grip power, and walking speed were significantly lower in new- versus non-OP groups (Table 1). Grip power and walking speed are criteria used to diagnose sarcopenia, and the percentage of subjects exhibiting sarcopenia newly present at the third survey was significantly higher in new OP than non-OP cases (Table 1).

We then analyzed levels of IGF1 and various metabolites in sera collected at the second survey and compared those levels between new OP and non-OP cases (Table 2). Levels of some metabolites (shown in Table 2) differed significantly between new OP and non-OP subjects.

Age is one of the most common risk factors for osteoporosis development, and age indeed differed significantly between new OP and non-OP cases (Table 1). Thus, the parameters for which levels were significantly different between new OP and non-OP cases were adjusted for age (Table 3). Moreover, female gender and BMI were considered potential risk factors, and indeed, percentages associated with female gender and BMI values differed significantly between new OP and non-OP cases (Table 1). Thus, after adjusting parameters significantly associated with new osteoporosis development for age, we further adjusted them for age, sex, and BMI (Table 3). After adjustment for age, sex, and BMI, we found that relatively high Gly levels remained significantly associated with new OP cases (odds ratio: 1.003; 95% confidence interval: 1.000–1.006; *p* = 0.021) (Table 3).

### 2.2. Low Serum Taurine Levels Are Significantly Associated with New Sarcopenia Development

Among participants, 492 did not satisfy the sarcopenia criteria at the second survey, while at the third survey, 39 (14 males and 25 females) of that 492 did. We analyzed patient characteristics at the second survey and compared them between new sarcopenia (new SP) and non-sarcopenia (non-SP) subjects, (Table 4). Age was significantly higher and BMI significantly lower in the new SP group compared to non-SP group (Table 4). Moreover, BMD, grip power, and walking speed were all significantly lower in the new SP group relative to the non-SP group (Table 4). Osteoporosis prevalence at the second and third surveys was significantly higher in new SP subjects (Table 4), suggesting an association with sarcopenia.

Subsequently, we analyzed levels of IGF1 and various metabolites in sera collected at the second survey. Metabolites whose levels significantly differed between new SP and non-SP groups are shown in Table 5. Serum IGF1 levels were significantly lower in the new SP group than the non-SP group (Table 5). Valine, leucine, and isoleucine, all branched-chain amino acids known to be abundant in skeletal muscle, were significantly lower in new SP cases compared to non-SP cases. Similarly, levels of taurine, also a skeletal muscle-related amino acid, were significantly lower in new SP cases compared to non-SP cases (Table 5).

We adjusted metabolites whose levels differed significantly between the new SP and non-SP groups by age (Table 6). We then adjusted factors significantly associated with new SP after adjustment with by age, sex, and BMI (Table 6). A logistic regression analysis after adjustment revealed that relatively reduced taurine levels (odds ratio: 0.991; 95% confidence interval: 0.982–0.999; *p* = 0.039) and existing osteoporosis (odds ratio: 3.118; 95% confidence interval: 1.215–7.997; *p* = 0.018) at the second survey predicted new sarcopenia development within the next four years (Table 6).

## 3. Discussion

Osteoporosis and sarcopenia are age-related diseases that are increasing in the number of cases and prevalence in developed countries. Both worsen ADL and QOL in the elderly. These conditions may be correlated and influence each other. In support of this idea, we found that among patients with newly developed osteoporosis, 15.2% also had newly developed sarcopenia. In contrast, 5.8% of subjects enrolled in the study who did not develop osteoporosis newly developed sarcopenia (Table 1). By contrast, 17.2% of patients with newly developed sarcopenia also newly developed osteoporosis, while 6.7% of subjects who did not newly develop sarcopenia newly developed osteoporosis (Table 4). Moreover, we found that elevated serum Gly levels predicted new osteoporosis development within four years but not the development of sarcopenia, while decreased serum taurine levels were associated with the new development of sarcopenia but not osteoporosis in a four-year period. Gly levels were higher in subjects with newly developed sarcopenia compared with those who did not, but differences were not statistically significant (non-SP vs. new SP: 494.3 ± 118.0 vs. 499.8 ± 138.6, respectively; *p* > 0.05). Moreover, taurine levels were significantly higher in subjects that newly developed osteoporosis relative to those who did not (non-OP vs. new OP: 252.0 ± 80.0 vs. 283.1 ± 126.1, respectively; *p* = 0.036). However, that difference also was not statistically significant after adjustment for age and BMI. Thus, although osteoporosis and sarcopenia are closely related diseases, methods to predict them likely differ overall.

Five subjects developed both osteoporosis and sarcopenia (the “both” group), but their Gly levels were comparable to those seen in individuals who developed osteoporosis but not sarcopenia (new OP alone) (both vs. new OP alone: 484.3 ± 93.7 vs. 561.0 ± 145.4, respectively; *p* > 0.05). Likewise, taurine levels were similar between the “both” group and subjects who developed sarcopenia but not osteoporosis (new SP alone) (both vs. new SP alone: 242.5 ± 49.7 vs. 221.3 ± 47.0, respectively; *p* > 0.05). In contrast, levels of the amino acid citrulline were significantly higher in the “both” group relative to new SP alone (57.8 ± 22.7 vs. 41.2 ± 10.3, respectively; *p* < 0.05). Citrulline supplementation reportedly promotes increased skeletal muscle mass [19], but at present, the mechanisms underlying increased citrulline levels in subjects who develop both osteoporosis and sarcopenia remain unclear.

Bone and muscle interact [20,21,22,23], and myokines produced from muscle regulate bone homeostasis [24]. For example, irisin, a myokine secreted from muscle with exercise, controls bone metabolism [25,26]. Extracellular matrix vesicles produced in and secreted by muscles reportedly regulate bone homeostasis [27]. Insulin-like growth factor 1 (IGF1), which is produced by osteoblasts and stored in extracellular matrix proteins in bones, is a bone-remodeling factor [28] that is released upon osteoclastic bone resorption and stimulates subsequent osteoblastic bone-forming activities. IGF1 also stimulates anabolic pathways and inhibits catabolic pathways in muscles [29]. Previously, we demonstrated that reduced IGF1 levels promoted decreased muscle mass in adult mice [30]. Indeed, here we showed lower serum IGF1 levels in the new- versus non-SP cases (Table 5). Transforming growth factor beta 1 (TGFβ1) is also a bone-remodeling factor released from the bone matrix [31]. TGFβ1 is activated by osteoclastic activity and stimulates bone formation by osteoblasts [31]. On the other hand, TGFβ1 activity promotes muscle atrophy [32]. Thus, the activities of bone and muscle are tightly linked, and the disruption of that interaction could promote the development of both osteoporosis and sarcopenia. Indeed, some subjects assessed here developed osteoporosis and sarcopenia concomitantly, although we showed that factors predictive of these conditions differ.

Several studies combining cohort and metabolomic analysis have been reported. For example, gut microbiome α-diversity is reportedly explained by a subset of 40 plasma metabolites in two independent human cohort blood samples [33]. Metabolite profiles are also reported to predict future diabetes [34]. Low plasma lysophosphatidylcholine (LPC) 18:2 is reportedly a predictor of decline in gait speed in older adults [35]. Here, although small sample size was a limitation of our study, we combined comprehensive metabolome analysis with a longitudinal cohort study to show for the first time that Gly and taurine levels in sera are predictors of osteoporosis and sarcopenia, respectively.

Glycine (Gly) is a non-essential amino acid, and increased glycine levels are reportedly seen in male idiopathic osteoporosis patients [36]. Moreover, in humans, supplementation with six amino acids including glycine is associated with a higher BMD in the spine and forearm [37]. We found that Gly levels were significantly and negatively correlated with bone mineral density (BMD) at either the lumbar spine L2–4 levels or the femoral neck (*r* = −0.1818 (*p* < 0.05) or −0.2134 (*p* < 0.05), respectively) at the second survey, supporting the idea that increased Gly levels are associated with decreased BMD. Gly is the most abundant amino acid residue in collagen [38], and collagen degradation products such as collagen cross-linked N-telopeptide (NTx) are frequently detected at high levels in the blood and urine of osteoporosis patients. Taurine is a non-essential amino acid synthesized from methionine and cysteine by an cysteine sulfinic acid decarboxylase (CSD). Older adults with sarcopenia reportedly exhibit higher levels of several amino acids in serum, including taurine [39]. Sarcopenia is known to be associated with undernutrition [40], and nutritional supplementation with taurine reportedly counteracts the development and progression of sarcopenia in human subjects [41]. We found that taurine levels were positively correlated with walk speed, one of the criteria used to assess sarcopenia, but that association was not statistically significant (*r* = 0.0378 and *p* > 0.05). The mechanisms underlying the changes in the serum levels of these amino acid levels before osteoporosis or sarcopenia development remain unclear; however, we conclude that the monitoring of these levels in individuals could be predictive of musculoskeletal disorders.

## 4. Materials and Methods

### 4.1. Subjects

This study was conducted using ROAD study cohorts. Subjects were enrolled from resident registration listings in three communities with different characteristics: 1350 subjects were from an urban region in Itabashi, Tokyo; 864 were from a mountainous region in Hidakagawa, Wakayama; and 826 were from a coastal region in Taiji, Wakayama. After the baseline study, the second survey was conducted in the same communities from 2008 to 2010 [42], and the third survey was performed from 2012 to 2013. In the second and third surveys, all examinations for osteoporosis (OP), sarcopenia (SP), and frailty were performed only in cohorts from mountainous and coastal regions. Therefore, here, 1551 participants (521 men and 1030 women) were enrolled from those two cohorts. Blood samples were collected from these subjects in the second survey and stocked. Additionally, from among the 1551 participants, those aged ≥60 years were selected based on the AWGS criteria for sarcopenia [9]. Consequently, 1083 (372 men and 711 women; mean age: 72.1 (SD: 7.4) years (72.7 (SD: 7.5) years for men; 71.7 (SD: 7.3) years for women)) participants were recruited as eligible subjects for the follow-up of sarcopenia. The third survey was carried out in the same subjects between 2012 and 2013, and 767 individuals (253 men and 514 women) participated in both the second and third surveys of the ROAD study among 1083 participants and completed assessment for both OP and SP. Both osteoporosis and sarcopenia were evaluated at the second and third surveys, and blood samples were collected and stored from participants from the mountainous region of Hidakagawa. We analyzed 729 participants (254 men and 475 women) aged 68.7 ± 11.1 years (69.3 ± 10.7 in men and 68.4 ± 11.3 in women) from Hidakagawa, and all completed the second and third follow-up surveys and provided blood samples. Written informed consent were provided by all participants, and the study was conducted with the approval of the ethics committees of the University of Tokyo (Nos. 1264 and 1326) and the University of Wakayama Medical University (No. 373). All procedures were conducted in accordance with the ethical standards as described in the 1964 Declaration of Helsinki and its later amendments.

Osteoporosis and sarcopenia were defined as described in [11]. Briefly, osteoporosis was defined based on bone mineral density (BMD) values, and subjects who exhibited a BMD T score lower than −2.5 SD were judged to be osteoporotic. Sarcopenia was defined based on the criteria of the AWGS [9,11]. New cases of either condition were defined as appearing in subjects who did not meet the osteoporosis or sarcopenia criteria at the second survey but satisfied those criteria at the third survey.

### 4.2. IGF1 ELISA

Serum IGF1 levels were measured by ELISA according to the manufacturer’s protocol (MG100, R&D Systems, Minneapolis, MN, USA) using a multiple plate analyzer (Powerscan HT, DS Pharma Biomedical, Osaka, Japan).

### 4.3. Metabolite Extraction from Serum

Frozen serum samples were thawed, and 40 µL aliquots were added to 360 µL of methanol containing internal standards (20 µmol/L each of methionine sulfone and D-camphor-10-sulfonic acid). Solutions were mixed well, and then both 400 µL of chloroform and 160 µL of Milli-Q water were added, followed by centrifugation at 10,000× *g* for 3 min at 4 °C. Then, 300 µL of the upper layer were transferred to a 5-kDa-cutoff filter (Human Metabolome Technologies, Tsuruoka, Japan) to remove proteins. The filtrate was dried using a centrifuge concentrator and reconstituted with 40 µL of Milli-Q water containing reference compounds (200 µmol/L each of 3-aminopyrrolidine and trimesic acid) prior to CE-TOFMS analysis.

### 4.4. Metabolome Analysis by CE-TOFMS

All CE-TOFMS analyses were performed using an Agilent 1600 Capillary Electrophoresis system (Agilent technologies, Waldbronn, Germany) equipped with an Agilent 6210 TOF LC/MS system (Agilent technologies, Santa Clara, CA, USA), as described in [5]. For anionic metabolite analysis, the ESI sprayer was replaced with a platinum needle rather than the original stainless steel needle [43]. Cationic metabolites were separated through a fused-silica capillary (50 µm i.d. × 100 cm) filled with 1 mol/L formic acid as an electrolyte [44] and methanol/water (50%, *v*/*v*) containing 0.1 µmol/L hexakis (2,2-difluoroethoxy) phosphazene was delivered as a sheath liquid at a rate of 10 µL/min. Capillary temperature was maintained at 20 °C.

Sample solution was injected at 5 kPa for 3 sec, and the separation voltage was set at 30 kV. ESI-TOFMS was conducted in positive ion mode, and capillary, fragmentor, skimmer, and Oct RF voltages were set at 4000, 75, 50, and 125 V, respectively. Nebulizer gas pressure was configured at 7 psig, and heated nitrogen gas (300 °C) was supplied at a rate of 10 L/min. Anionic metabolites were separated through a COSMO (+) capillary (50 µm i.d. × 105 cm, Nacalai Tesque, Kyoto, Japan) filled with 50 mmol/L ammonium acetate (pH 8.5) as an electrolyte [43], and ammonium acetate (5 mmol/L) in 50% (*v*/*v*) methanol/water containing 0.1 µmol/L hexakis (2,2-difluoroethoxy) phosphazene was delivered as a sheath liquid at a rate of 10 µL/min. A sample solution was injected at 5 kPa for 30 s, and −30 kV was applied to promote sample separation. ESI-TOFMS was conducted in negative ion mode, and capillary, fragmentor, skimmer, and Oct RF voltages were set at 3500, 100, 50, and 200 V, respectively. Other conditions were identical for cationic metabolite analysis.

In both modes, an automatic recalibration function was used to correct for the analytical variation of exact masses for each run, as described in [45]. Exact mass data were acquired at a rate of 1.5 cycles/s over a 50–1000 *m*/*z* range. Raw data were processed using our proprietary automatic integration software (MasterHands) [46,47]. Each peak was identified by matching *m*/*z* values and the normalized migration times of corresponding authentic standard compounds. We assessed 238 anionic and 272 cationic metabolites, and we detected a total 80 (as listed in Appendix A). Metabolite levels are reported as μM.

### 4.5. Statistical Analysis

The statistical significance of indicated parameters was analyzed between subjects who were not diagnosed with osteoporosis at both the second and third surveys (non-osteoporosis) and those who were not diagnosed with osteoporosis at the second survey but newly developed osteoporosis after the second survey (new osteoporosis). Similarly, the statistical significance of various parameters was analyzed between subjects who were not diagnosed with sarcopenia at both second and third surveys (non-sarcopenia) and those who were not diagnosed with sarcopenia at the second survey but newly developed sarcopenia after that survey (new sarcopenia). All statistical analysis was performed using the STATA software (STATA Corp., College Station, TX, USA).

## Figures and Tables

**Table 1 metabolites-11-00278-t001:** Comparison of characteristics between non- and new-osteoporosis subjects at the second survey.

	Non-Osteoporosis	New Osteoporosis	*p* Value
Age (years)	65.2 ± 10.8	69.8 ± 8.0	0.014
Female sex (%)	60.0	80.0	0.0181
Height (cm)	156.0 ± 8.8	150.6 ± 8.0	0.0004
Weight (kg)	57.1 ± 9.4	50.0 ± 6.6	<0.0001
BMI (kg/cm^2^)	23.4 ± 3.0	22.0 ± 2.5	0.0076
Waist (cm)	82.6 ± 8.6	79.2 ± 9.9	0.0295
BMD (L2–4) (g/cm^2^)	0.993 ± 0.172	0.818 ± 0.147	<0.0001
BMD (femoral neck) (g/cm^2^)	0.701 ± 0.121	0.562 ± 0.355	<0.0001
Grip strength (maximum) (kg)	32.1 ± 9.2	26.4 ± 6.8	0.0004
Usual walking speed (m/s)	1.13 ± 0.24	1.09 ± 0.23	0.3843
Sarcopenia at the second survey (%)	2.4	5.7	0.2314
New sarcopenia at the third survey (%)	5.8	15.2	0.0353

Non-osteoporosis: subjects who were not diagnosed with osteoporosis at both second and third surveys; new osteoporosis: subjects who were not diagnosed with osteoporosis at the second survey but newly developed osteoporosis after the second survey. Statistical significance was evaluated by a *t* test.

**Table 2 metabolites-11-00278-t002:** Levels of IGF1 and metabolites in non- and new-osteoporosis subjects at the second survey.

	Non-Osteoporosis	New Osteoporosis	*p* Value
IGF1 (ng/mL)	86.0 ± 31.0	71.2 ± 12.8	0.0052
5-Oxoproline	81.6 ± 39.7	96.0 ± 47.7	0.0422
2-Oxoglutarate	13.1 ± 8.6	8.6 ± 6.5	0.0027
cis-Aconitate	3.61 ± 0.87	3.98 ± 0.78	0.0154
Gly	486.0 ± 113.6	550.1 ± 140.8	0.0017
3-Aminoisobutyrate	2.10 ± 2.16	3.07 ± 2.02	0.0109
Taurine	252.0 ± 80.1	283.1 ± 126.1	0.0364
Pipecolate	2.60 ± 3.09	4.99 ± 15.85	0.0099

Non-osteoporosis: subjects who were not diagnosed with osteoporosis at both second and third surveys; new osteoporosis: subjects who were not diagnosed with osteoporosis at the second survey but newly developed osteoporosis after the second survey. Statistical significance was evaluated by a *t* test. Statistical significance was evaluated by *t* test.

**Table 3 metabolites-11-00278-t003:** Logistic regression model for new osteoporosis development.

	*p* Value after Adjustment for Age	*p* Value after Adjustment for Age, Sex, and BMI	Odds Ratio	95% Confidence Interval
BMD (L2–4) (g/cm^2^)	<0.001	<0.001	0.0004	9.72 × 10^−6^–0.161
BMD (femoral neck) (g/cm^2^)	<0.001	<0.001	3.53 × 10^−8^	5.47 × 10^−11^–0.00002
Grip strength (maximum) (kg)	0.003	0.174		
IGF1	0.088			
5-Oxoproline	0.07			
2-Oxoglutarate	0.003	0.055		
cis-Aconitate	0.088			
Gly	0.001	0.021	1.003	1.000–1.006
3-Aminoisobutyrate	0.066			
Taurine	0.038	0.127		
Pipecolate	0.095			

Statistical significance was evaluated by logistic regression analysis.

**Table 4 metabolites-11-00278-t004:** Comparison of characteristics between non- and new-sarcopenia subjects at the second survey.

	Non-Sarcopenia	New Sarcopenia	*p* Value
Age (years)	64.9 ± 10.5	75.9 ± 5.7	<0.0001
Female sex (%)	65.1	64.1	0.8983
Height (cm)	155.2 ± 8.9	151.7 ± 8.8	0.0197
Weight (kg)	56.5 ± 9.5	48.6 ± 7.1	<0.0001
BMI (kg/cm^2^)	23.4 ± 3.0	21.1 ± 2.4	<0.0001
Waist (cm)	82.5 ± 8.5	77.4 ± 8.9	0.0004
BMD (L2–4) (g/cm^2^)	0.959 ± 0.189	0.877 ± 0.228	0.0106
BMD (femoral neck) (g/cm^2^)	0.685 ± 0.125	0.605 ± 0.134	0.0002
Grip strength (maximum) (kg)	31.6 ± 9.2	26.1 ± 7.0	0.0003
Usual walking speed (m/s)	1.13 ± 0.23	0.98 ± 0.15	<0.0001
Osteoporosis at the second survey (%)	10.6	46.2	<0.0001
New osteoporosis at the third survey (%)	6.7	17.2	0.0353

Non-sarcopenia: subjects who were not diagnosed with sarcopenia at both second and third surveys; new sarcopenia: subjects who were not diagnosed with sarcopenia at the second survey but newly developed sarcopenia after the second survey. Statistical significance was evaluated by a *t* test.

**Table 5 metabolites-11-00278-t005:** Levels of IGF1 and metabolites in non- and new-sarcopenia subjects at the second survey.

	Non-Sarcopenia	New Sarcopenia	*p* Value
IGF1 (ng/mL)	85.0 ± 30.8	71.5 ± 24.5	0.0077
2-Hydroxybutyrate	27.5 ± 11.6	23.5 ± 9.2	0.0376
4-Methyl2oxopentanoate	50.3 ± 14.0	44.0 ± 13.8	0.0071
2-AB	17.1 ± 5.6	14.9 ± 5.3	0.0221
Val	350.7 ± 72.1	326.0 ± 70.4	0.0040
Ile	90.9 ± 26.9	82.0 ± 21.4	0.0448
Leu	196.5 ± 49.6	175.4 ± 46.0	0.0107
Taurine	257.6 ± 86.5	227.5 ± 46.3	0.0328
Trp	72.9 ± 13.6	66.9 ± 14.1	0.0079
Cystine	42.1 ± 8.6	38.5 ± 7.8	0.0122

Non-sarcopenia: subjects who were not diagnosed with sarcopenia at both second and third surveys; new sarcopenia, subjects who were not diagnosed with sarcopenia at the second survey but newly developed sarcopenia after the second survey. Statistical significance was evaluated by a *t* test.

**Table 6 metabolites-11-00278-t006:** Logistic regression model for new sarcopenia development.

	*p* Value after Adjustment for Age	*p* Value after Adjustment for Age, Sex, and BMI	Odds Ratio	95% Confidence Interval
BMD (L2–4) (g/cm^2^)	0.076			
BMD (femoral neck) (g/cm^2^)	0.066			
Grip strength (maximum) (kg)	0.071			
Usual walking speed (m/s)	0.263			
Osteoporosis at the second survey (%)	<0.001	0.018	3.118	1.215–7.997
IGF1	0.915			
2-Hydroxybutyrate	0.093			
4-Methyl2oxopentanoate	0.093			
2-AB	0.077			
Val	0.06			
Ile	0.06			
Leu	0.074			
Taurine	0.038	0.039	0.991	0.982–0.999
Trp	0.102			
Cystine	0.03	0.108		

Statistical significance was evaluated by logistic regression analysis.

## Data Availability

All data are contained within the article.

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
