# Peer review of "A Metabolomic Profile Predictive of New Osteoporosis or Sarcopenia Development"

_metabolites, 2021, doi:10.3390/metabo11050278_

Round 1
Reviewer 1 Report
metabolites-1191112
Metabolomic profile predictive of new osteoporosis or sarcopenia development
The manuscript provides a revised version of the manuscript “Metabolomic profile predictive of new osteoporosis or sarcopenia development”. The manuscript has been improved considerably and my comments have been adequately addressed.
Author Response
- The manuscript provides a revised version of the manuscript “Metabolomic profile predictive of new osteoporosis or sarcopenia development”. The manuscript has been improved considerably and my comments have been adequately addressed.
Reply: We thank you for valuable comments and appreciate this response.
Reviewer 2 Report
The authors have addressed most of my concerns during last round of review. However, I still have strong concerns for the conclusion that Gly and taurine as predictors because of the small sample size and the values of the Odds ratio and 95% confidence interval. Therefore, I suggest the authors to weaken the conclusion throughout the manuscript including title and abstracct. Maybe it is better to state that the changes of Gly and taurine are associated with the disease rather than state that they are predictors.
Author Response
- The authors have addressed most of my concerns during last round of review. However, I still have strong concerns for the conclusion that Gly and taurine as predictors because of the small sample size and the values of the Odds ratio and 95% confidence interval. Therefore, I suggest the authors to weaken the conclusion throughout the manuscript including title and abstracct. Maybe it is better to state that the changes of Gly and taurine are associated with the disease rather than state that they are predictors.
Reply: We thank the reviewer for valuable comments that helped us improve our manuscript. We understand that our sample size is limited, and that values of the odds ratio and 95% interval are significant but small. However, ours was a longitudinal prospective study, and new osteoporosis or sarcopenia development was evident at the third survey in subjects who exhibited lower Gly or higher taurine, respectively, at the second survey during four-year follow-up. Moreover, logistic regression analysis demonstrated that low Gly or high taurine significantly predicted new osteoporosis or new sarcopenia development, respectively, at the third survey, even after adjustment for age, sex and BMI. Thus we feel that our original usage of the word “predict” is appropriate. However, we agree with your comments regarding small sample size and odds ratio and now state that in the Abstract, saying, “Although sample size and odds ratios are small….”, and state that low Gly or high taurine are “predictive” rather than “predictor” in the title and abstract. We also change the title to “A Metabolomic Profile Predictive of New Osteoporosis or Sarcopenia Development”.